# Discovering of Genomic Variations Associated to Growth Traits by GWAS in Braunvieh Cattle

**DOI:** 10.3390/genes12111666

**Published:** 2021-10-22

**Authors:** José Luis Zepeda-Batista, Rafael Núñez-Domínguez, Rodolfo Ramírez-Valverde, Francisco Joel Jahuey-Martínez, Jessica Beatriz Herrera-Ojeda, Gaspar Manuel Parra-Bracamonte

**Affiliations:** 1Facultad de Medicina Veterinaria y Zootecnia, Universidad de Colima, Kilometro 40 Autopista Colima-Manzanillo, Tecomán 28100, Colima, Mexico; jzepedab@gmail.com; 2Departamento de Zootecnia, Posgrado en Producción Animal, Universidad Autónoma Chapingo, Km. 38.5 Carretera México-Texcoco, Chapingo 56230, Texcoco, Mexico; rafael.nunez@correo.chapingo.mx (R.N.-D.); rrv33@hotmail.com (R.R.-V.); 3Facultad de Zootecnia y Ecologa, Universidad Autónoma de Chihuahua, Periférico Francisco R. Almada, Km 1, Chihuahua 33820, Chihuahua, Mexico; fco_jahuey@hotmail.com; 4Departamento de Ciencias Básicas, Instituto Tecnológico del Valle de Morelia, Instituto Tecnológico Nacional, Morelia 58100, Michoacán, Mexico; ysika_ho@hotmail.com; 5Centro de Biotecnología Genómica, Instituto Politécnico Nacional, Boulevard del Maestro S/N esq. Elías Piña, Col. Narciso Mendoza, Ciudad Reynosa 88710, Tamaulipas, Mexico

**Keywords:** association, candidate gene, growth, quantitative trait loci, single nucleotide polymorphism

## Abstract

A genome-wide association study (GWAS) was performed to elucidate genetic architecture of growth traits in Braunvieh cattle. Methods: The study included 300 genotyped animals by the GeneSeek^®^ Genomic Profiler Bovine LDv.4 panel; after quality control, 22,734 SNP and 276 animals were maintained in the analysis. The examined phenotypic data considered birth (BW), weaning (WW), and yearling weights. The association analysis was performed using the principal components method via the egscore function of the GenABEL version 1.8-0 package in the R environment. The marker rs133262280 located in BTA 22 was associated with BW, and two SNPs were associated with WW, rs43668789 (BTA 11) and rs136155567 (BTA 27). New QTL associated with these liveweight traits and four positional and functional candidate genes potentially involved in variations of the analyzed traits were identified. The most important genes in these genomic regions were *MCM2* (minichromosome maintenance complex component 2), *TPRA1* (transmembrane protein adipocyte associated 1), *GALM* (*galactose mutarotase*), and *NRG1* (neuregulin 1), related to embryonic cleavage, bone and tissue growth, cell adhesion, and organic development. This study is the first to present a GWAS conducted in Braunvieh cattle in Mexico providing evidence for genetic architecture of assessed growth traits. Further specific analysis of found associated genes and regions will clarify its contribution to the genetic basis of growth-related traits.

## 1. Introduction

The identification of causal genetic variability is one of the main goals in the genetic improvement of cattle. Commonly, liveweight traits are used as the primary selection criterion in cow-calf production systems in Mexico [1]. Usually, farms use these traits as efficiency and meat potential production indicators, and they are used for genetic evaluations in most of the registered breeds [2].

Braunvieh is a worldwide cattle breed used in the beef industry that has been used in both specialized beef and dual-purpose production herds [3,4]. Due to its initial dual-purpose origin, most of the available information about the Braunvieh deals with dairy production traits. However, during the last 15 years, Braunvieh cattle have been selected and genetically improved for beef production traits [4,5]. In Mexico, Braunvieh is one of the breeds most utilized for the beef production industry either as purebred or in crosses with *Bos indicus* cattle [3,6]. However, despite its extensive use, there is scarce information on the breed’s productive performance, and the available information is mainly related to growth traits coming from genetic evaluations or isolated studies [5,6,7]. The selection, management, and genetic improvement programs of the Braunvieh cattle could be enhanced using high-throughput genotyping technologies.

The use of microarrays of thousands of SNP markers in genome-wide association studies (GWAS) has allowed discovering the genetic basis of complex traits and diseases by detecting genotype–phenotype associations in a group of individuals [8]. GWAS approaches have confirmed many QTL for growth traits in beef and crossbred cattle [1,9,10], some of which have been used as the basis for the search for specific causal variation [11] and a better understanding of the genetic architecture of these complex traits. Many of these QTL, genomic regions and genes, affecting production traits in beef cattle have been reported [1,12,13,14], but most of the association studies focus on specialized beef breeds and only a few studies have been implemented in breeds such as Braunvieh [15,16]. The present study is aimed at performing a GWAS to identify QTLs and candidate genes related to liveweight traits in a Braunvieh cattle population.

## 2. Materials and Methods

### 2.1. Population and Phenotypic Data

Hair follicle samples from 236 females and 64 males registered in the Mexican Braunvieh Cattle Association database were collected. The cattle were born between 2000 and 2015. This population came from herds located in the east, west, and central highlands of Mexico. Herds from west and east were raised under extensive production systems, while central highlands herds were under intensive regimens. The sampled population’s genetic background included Austrian, Swiss, Canadian, American, and Mexican animals. Phenotypic data were provided by the breeding association and included records of birth weight (BW, kg), weaning weight (WW, kg), yearling weight (YW, kg). Weaning and yearling weights were adjusted to perform the GWAS analysis.

### 2.2. Genotyping and Quality Control

The animals were genotyped using 30,125 SNP markers from the GeneSeek^®^ Genomic Profiler Bovine LDv.4 panel (Neogen Corp., Lincoln, NE, USA). Before association analysis, the genotypic data quality was verified using the SNPQC program [17]. The genotypes were considered successful if they presented a GenCall value greater than 0.50, and all SNPs with lower values were discarded (*n* = 1623). Those SNPs that were monomorphic (*n* = 3604), presented call rates of less than 90% (*n* = 1290) or minor allele frequencies < 0.01 (*n* = 1325), or deviated from the Hardy–Weinberg equilibrium according to Fisher’s exact test and exhibited *p*-values >1 × 10^−15^ (*n* = 0) were also eliminated. Additionally, SNPs with unknown coordinates in the assembly of the bovine genome UMD v3.1 [18] (*n* = 1484) and SNPs that were not located on autosomal chromosomes (*n* = 1820) were discarded.

Samples were also eliminated if they exhibited call rates of less than 80% (*n* = 0) or levels of heterozygosity (HE) above 3 SD (*n* = 1), considering that the mean and SD of the observed HE were 0.32 and 0.019, respectively. A Pearson correlation was computed for detecting potentially duplicate samples, considering a maximum of *r* = 0.98, according to their genotype information obtaining an average of *r* = 0.817 and minimum and maximum values of 0.66 and 0.90, respectively. A total of 22,734 SNPs and 276 samples passed the quality control procedure and were retained for further analysis. Quality control and subsequent analyses were performed in the R environment.

### 2.3. Population Structure and Association Analysis

Population structure was analyzed, calculating first a genomic relationships matrix using the information on genotypes [19], in addition to performing a singular value decomposition and a principal components (PC) analysis.

The PC analysis indicated that the first two PCs explained 28.6% of the variance in the data. A multidimensional scaling analysis confirmed this structure (Figure 1). Therefore, the genome-wide association analysis was performed using the PC method proposed by Price et al. [20]. For this analysis, the egscore function from the GenABEL package [21] was employed. This function accounts for population stratification and uses the genomic kinship matrix to derive axes of genetic variation, and then both phenotypes and genotypes are adjusted onto these axes.

A linear model for each trait was fitted, including the first two PC as covariates. For the analysis of BW, the model also included the contemporary group (CG) and the linear and quadratic effects of cow age at the birth and weaning of her calf. The CG included herd, sex, year, and calving season. The statistical model used to adjust the other traits only included the CG and the PCs as covariates; cow age was excluded because it was not significant in the previous analysis. Finally, the association between corrected genotypes and phenotypes was assessed via correlation. *p*-values were obtained by calculating the square of the correlation multiplied by (N-K-1), where N was the number of genotyped individuals, and K was the number of PCs.

Minimum allele frequencies, allele substitution effect (β), and percentage of phenotypic variance explained by the SNP were estimated. SNP with *p*-values < 5 × 10^−5^ were considered significantly associated with studied traits. The proportion of phenotypic variance explained by the SNPs was estimated by dividing the *x*^2^ value for a df by the number of individuals used to analyze each SNP marker, followed by multiplication by 100. All described analyses and estimations were performed using the GenABEL package [21].

### 2.4. Analysis of Genomic Regions with Significant SNPs

The closest genes to significant markers and those located within a 250 kb window on both SNP location sides were identified. The list of genes was obtained using the snp2gene.LD function from the Postgwas package [22]. Distance between SNPs and genes was calculated as the difference between the marker position and the beginning or end of the gene, according to coordinates from bovine genome assembly UMD v3.1. Gene functions were investigated in the UniProt database [23].

Annotations from humans or mice were used when there was no information on the genes in cattle. Genes were considered functional and positional candidates if they were biologically related to the trait under study, supported by experimental evidence in the literature. Finally, we determined whether significant SNPs mapped against QTLs previously associated with growth-related traits such as BW, carcass, and reproduction traits, deposited in the cattle AnimalQTLdb [24]. For this purpose, SNP positions according to the Btau4.6 genome sequence were used because many of the previously reported QTLs had no well-defined positions in the bovine genome assembly UMD v3.1.

## 3. Results

A total of 30,125 SNP markers from the GeneSeek^®^ Genomic Profiler Bovine LD v4 microarray panel (Neogen Corp. Lincoln, NE, USA) were used for association with live weight traits of Braunvieh cattle. On average, 1004 SNP markers were evaluated in each BTA. *Bos taurus* chromosomes 1 and 27 exhibited the highest (1602) and lowest (512) SNP numbers. The average distance between adjacent SNP was 87,641 bp, the minimum distance (0 bp) between adjacent SNP were found on BTA 1, 6, 7, 12, 17, 18, 22, 25, 26, 28, and 29, while the maximum distance (1,962,000 bp) was found on BTA 6. Table 1 show the descriptive statistics for each trait.

Figure 2 shows quantile–quantile plots for each GWAS analysis performed. According to the significance threshold considered (*p* < 5 × 10^−5^), 3 SNP were associated with the studied growth traits. Two SNP was associated with BW and one SNP was associated to WW. The rs133262280 located in BTA22 was associated with BW, showing an allelic substitution effect of 0.320 ± 0.02 kg. The rs43668789 and rs136155567, located in BTA11 and 27, respectively, were associated with WW. These markers showed allelic substitution effects of −9.590 ± 0.25 and 1.110 ± 0.72 kg, respectively (Table 2, Figure 3).

Table 3, presents QTLs identified near the associated SNPs for BW and WW. Figure 3, shows the Manhattan plots in which the −log10 transformations of the *p*-values are plotted for each GWAS. 

Table 4 and Table 5, show complete descriptions of genes close to the SNP associated with BW and WW of Braunvieh cattle, including the identifier number and exact location identified in this study.

## 4. Discussion

The inclusion of the population’s genetic structure and fixed effect into the analysis model allowed the better fitting of the GWAS model for all traits, as showed by quantile–quantile plots (Figure 2). This genetic structure was expected because tested herds presented different selection criteria, and perhaps, ancestors from the imported genetic material (i.e., semen, sires). Stratification results could include extensive use of sires or semen that breeders usually choose in their genetic improvement programs. Some studies [30,31] have used subdivisions to estimate QTLs using genome-wide association studies (GWAS). Smitz et al. [32] concluded that the stratification in the studied populations needs to be considered in genetic improvement programs to conserve those populations “genetic health”. Jemaa et al. [33] indicated that some QTLs found in GWAS could not be present in all the studied animals due to the population’s stratification.

Birth weight in Braunvieh cattle represents an important trait to consider in the genetic improvement programs due to its association with calving difficulty in young heifers, especially when the Braunvieh is used as a sire for smaller-size breeds [34]. In the present study, the rs133262280 was identified as the only marker associated with BW, located at 60.7 Mb of BTA 22. This SNP showed an allelic substitution effect of 0.320 kg, explaining 0.1% of the phenotypic variance of BW. Genes located closer to this SNP included *CHCHD6* (coiled-coil helix coiled-coil helix domain-containing 6), *MCM2* (mini-chromosome maintenance complex component 2), *PLXNA1* (plexin A1), *PODXL2* (podocalyxin like 2), *TPRA1* (transmembrane protein adipocyte associated 1), and uncharacterized *LOC10105309* (Table 4). The most important genes identified in this region were *MCM2* and *TPRA1*. The *MCM2* gene is located at 177.6 kb and *TPRA1* at 160.1 kb; both genes are upstream of the rs133262280 SNP. *MCM2* acts as a component of the MCM2-7 complex (MCM complex) which is the putative replicative helicase essential for “once per cell cycle” DNA replication initiation and elongation in eukaryotic cells [31]. Additionally, it plays a role in cell division and apoptosis [35]. Gao et al. [36] reported MCM2 protein expression in the cochlea of rats and guinea pigs slightly increase the apoptosis rate of the cells without any changes in proliferation or cell cycle. Recently, Khan et al. [37] found by a transcriptomic analysis that supplementation with folic acid in perinatal Holstein cows significantly increases the expression of the *MCM2* gene.

The other associated gene with biological importance was the *TPRA1* gene belonging to the G protein-coupled receptor (GPCR) family. Functions related to this gene include regulating early embryonic cleavage and enhancing the hedgehog signaling pathway [38,39]. Several studies have highlighted its importance in pre- and perinatal tissue development in mice. Aki et al. [38] determined that the TPRA1 gene influenced the hedgehog signaling pathway, which plays an essential role in vertebrate embryonic tissue patterning of many developing organs, showing differences of around 50% in the signaling levels comparing homozygotes and heterozygotes animals.

This evidence suggests that *MCM2* and *TPRA1* could participate in the early stages of cattle development and, therefore, influence BW. There were no quantitative trait loci previously located in this region, which could be a specific QTL of the studied population.

The present study identified two regions (Table 3, WW_rs43668789_11_21.3 and WW_rs136155567_27_27.0) previously reported by McClure et al. [29] as associated with weaning weight and calving ease in Angus cattle. Furthermore, Boichard et al. [28] and Buitenhuis et al. [26] reported associations between the identified regions in this study and conformation traits, explaining between 5.9 and 8.9 % of the structural soundness in ten European dairy cattle breeds. On the other hand, Sherman et al. [27] and Rolf et al. [14] reported associations with allele substitution effects between -0.319 to 2.199 kg for feeding traits like average daily gain and residual feed intake in Angus, Charolais, and Canadian beef hybrid cattle.

From the associated WW rs43668789 associated SNP, genes located closer or covering this SNP included *ARHGEF33* (Rho guanine nucleotide exchange factor 33), *CDKL4* (cyclin-dependent kinase-like 4), *DHX57* (DExH-box helicase 57), *GALM* (galactose mutarotase), *GEMIN6* (gem nuclear organelle associated protein 6), *LOC104973309* (ubiquitin-40S ribosomal protein S27a pseudogene), *LOC107132913* (uncharacterized *LOC107132913*), *LOC782845* (60S ribosomal protein L23a pseudogene), *MAP4K3* (mitogen-activated protein kinase 3), *MIR2284Z*-2 (microRNA 2284z-2), *MORN2* (MORN repeat containing 2), *SOS1* (SOS Ras/Rac guanine nucleotide exchange factor 1), and *SRSF7* (serine and arginine-rich splicing factor 7) (Table 5). Interestingly, the position of rs43668789 in intronic *ARHGEF33* gene need further attention. Although unreported effects of intronic regulation in this gene were found, some evidence indicates that transcriptional regulations by intronic SNPs is possible [40]. Moreover, intronic variation might carry deeper functional effects since they are related to different types of noncoding RNAs (ncRNAs) in genomes including miRNAs, siRNAs, piwi-interacting RNAs (piRNAs), long noncoding RNAs (lncRNAs), and small nucleolar RNAs (snoRNAs), and they are known to be located in the intron regions within genes [41].

The most important gene identified in this region was *GALM*. This gene is located 217.4 kb upstream of the rs43668789 and belongs to the proteins that convert the α-aldose to β-anomer. *GALM* is involved in the pathway hexose metabolism, which is part of carbohydrate metabolism [42]. McClure et al. [29] reported a positive association of *GALM* with the weaning weight in Angus cattle. Shin et al. [43] mentioned that the association between *GALM* and the weaning weight in Holstein and Hanwoo cattle lies in quantity and the quality of the calves’ milk consumption. Quantitative trait loci located in this region have been previously associated with weaning weight in Angus [29], conformation in dairy cattle breed [26,28], and residual feed intake in Canadian beef synthetic cattle [27].

The second marker associated with WW was rs136155567, located at 27.0 Mb of BTA 27, and its allele substitution effect was 1.110 kg which explains 1.1% of the phenotypic variance. Genes located closer to this SNP (±600 kb) included *LOC104976093* (uncharacterized LOC104976093) and *NRG1* (neuregulin 1) (Table 5). *NRG1* was the most important gene identified. This gene is located at 567.1 kb downstream of the rs136155567. It is considered the direct ligand for ERBB3 and ERBB4 tyrosine kinase receptors. The multiple isoforms perform diverse functions, such as inducing growth and differentiation of epithelial, glial, neuronal, and skeletal muscle cells, and influence motor and sensory neuron development [44,45]. In cattle, *NRG1* has been highly associated with organ development [46]. Zhao [47] mentioned that this gene could influence the weaning weight as an emerging regulator of prolactin secretion.

In general, the phenotypic variance explained by the SNPs identified in this study was marginal (1.39% on average). In growth trait studies, it is expected that most SNP markers will explain only a tiny proportion of the observed phenotypic variance due to the polygenic control over such traits and because individual genes only slightly influence a phenotype. However, consideration of SNPs’ sets that are significantly associated with each trait may allow a greater proportion of phenotypic variance to be explained. For example, the two SNPs associated with WW could explain 4.08% of the variance in that trait. It is important to note the lack of significant association for YW. Although not a highly strict threshold was chosen, the amount of data reduction from BW and WW to YW might be related to these outcomes. However, the present outcomes increase knowledge of the genetic architecture of growth traits important in beef cattle production.

## 5. Conclusions

In conclusion, in the present study, three SNPs were associated with the assessed growth traits of Braunvieh cattle. Two SNPs were located in intergenic regions, and one was located in an intronic region of the *ARHGEF33* gene. Additionally, evidence shows that some of the genes closer to the three identified SNPs markers are functionally related to growth through embryonic cleavage, bone and tissue growth, cell adhesion, and organ development. There were four candidate genes with potential associations with assessed live weight traits in Braunvieh cattle, including *MCM2*, *TPRA1*, *GALM*, and *NRG1*. Subsequent studies examining these genomic regions could lead to the identification of polymorphisms with potential uses in the marker-assisted selection, providing a deeper understanding of the genetic basis and genetic architecture of growth traits in cattle. This study represents the first study to describe a GWAS conducted in Braunvieh cattle in Mexico. Further analysis using the present information would allow to conduct assessments on the ontogeny and specific search of causative mutations for live weight traits. Furthermore, examining particular and general genic effects would indicate the possibility of including genomic information into current genetic evaluations.

## Figures and Tables

**Figure 1 genes-12-01666-f001:**
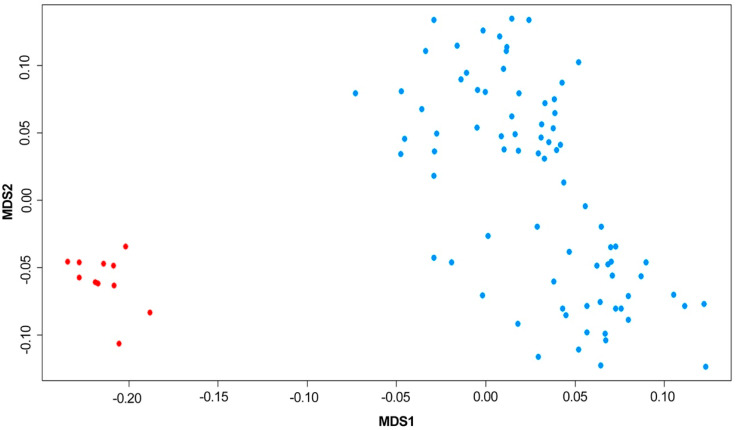
Muldimensional scaling analysis showing population genomic structure in the studied Braunvieh population.

**Figure 2 genes-12-01666-f002:**
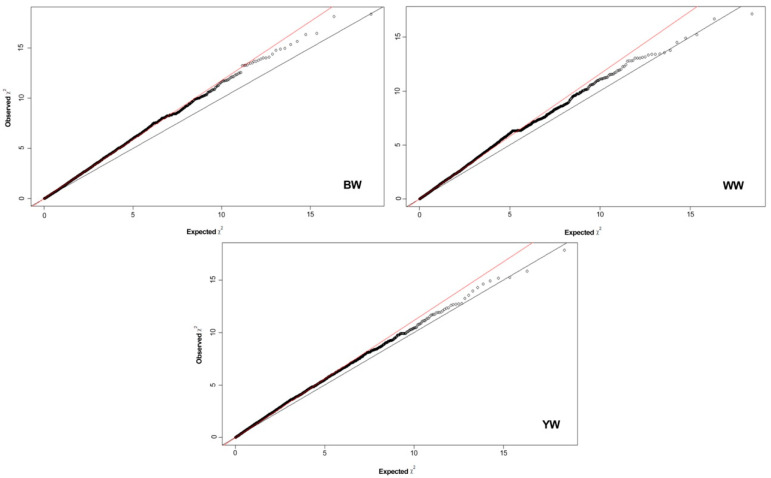
Quantile–quantile (QQ) plots for the genome-wide association study of birth (BW), weaning (WW) and yearling (YW) weight traits in Braunvieh cattle. The straight line in the QQ plots indicates the distribution of SNP markers under the null hypothesis, and the skew at the edge indicates that these markers are more strongly associated with the traits than would be expected by chance. BW = birth weight; WW = weaning weight; YW = yearling weight.

**Figure 3 genes-12-01666-f003:**
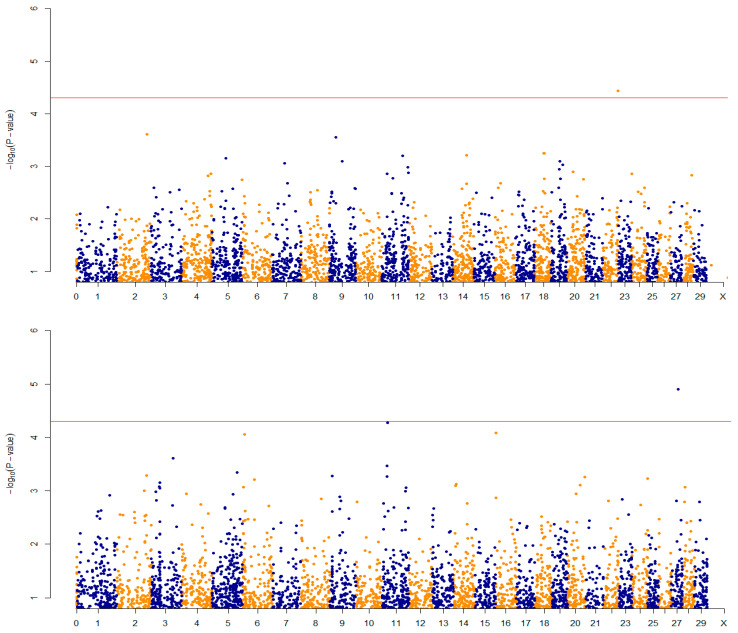
Manhattan plots of the *p*-values for the genome-wide association study of birth, weaning and yearling weights of Braunvieh cattle. The horizontal line indicates the significance threshold for significant associations (*p* < 5 × 10^−5^). Blue and orange differentiate chromosomes.

**Table 1 genes-12-01666-t001:** Descriptive statistics for studied live weight traits (kg) of Braunvieh cattle.

Trait ^1^	*N*	*n*(QC) ^2^	Mean	SD	Minimum	Maximum
BW	300	266	38.007	4.067	22	50
WW	300	263	212.399	27.426	128	308
YW	300	244	313.165	45.473	176	440

^1^ BW: birth weight; WW: weaning weight; YW = yearling weight; SD: Standard deviation; ^2^
*n*(QC) = *n* after quality control.

**Table 2 genes-12-01666-t002:** Parameters and statistics of SNP associated with liveweight traits of Braunvieh cattle.

Trait	SNP ID ^2^	BTA	UMD ^3^ bp	Btau4.6,^4^ bp	Allele	MAF ^5^	Β ^6^	SE	Var% ^7^	*p*-Value
^1^ BW	rs133262280	22	60,759,211	127,745,473	C/T	0.18	0.320	0.02	0.1	2.74 × 10^−5^
WW	rs43668789	11	21,312,462	22,502,811	C/T	0.17	−9.590	0.25	2.98	5.28e − 5
	rs136155567	27	27,056,807	29,944,194	A/G	0.20	1.110	0.72	1.1	1.27e − 5

^1^ BW = birth weight; WW = weaning weight; ^2^ ID = identification; ^3^ UMD version 3.1 [18]; ^4^ Elsik et al. [25]; ^5^ MAF = minimum allele frequency; ^6^ β = allele substitution effect; ^7^ Var% = phenotypic variance explained by the SNP.

**Table 3 genes-12-01666-t003:** Previously reported QTL1 found near the SNP associated with growth traits of Braunvieh cattle.

Trait_SNP ID ^2^_BTA_Mb	QTL	QTL ID	QTL in Btau4.6, ^3^ bp	QTL Reference
BW_rs133262280_22_60.7	—	—	—	—
WW_rs43668789_11_21.3	SOUND	3591	18,215,471–23,417,727	Buitenhuis et al. [26]
	RFI	5281	8,076,786–33,430,175	Sherman et al. [27]
	RANGLE	3447	16,291,959–80,096,141	Boichard et al. [28]
	WWTMM	10894	16,291,959–80,096,141	McClure et al. [29]
WW_rs136155567_27_27.0	BQ	3598	24,473,016–31,018,770	Buitenhuis et al. [26]
	SOUND	3594	24,473,016–31,018,770	Buitenhuis et al. [26]
	ADFI	21028	27,034,490–29,073,970	Rolf et al., [14]
	ADG	20979	27,034,490–29,073,970	Rolf et al., [14]
	RFI	21095	27,034,490–29,073,970	Rolf et al. [14]
	CALEASE	11259	21,801,052–31,012,980	McClure et al. [29]

ADG = average daily gain; ADFI = average daily feed intake; BQ = bone quality; CALEASE = calving ease; RFI = residual feed intake; RANG = rump angle; SOUND = structural soundness; WWTMM = weaning weight–maternal milk; ^2^ ID = identification; ^3^ Elsik et al. [25].

**Table 4 genes-12-01666-t004:** Genes close to the SNP rs133262280_22 associated with birth weight of Braunvieh cattle.

SNP_BTA	Gene in ±250 kb ^1^	Gene ID ^2^	Distance, ^3^ kb	Description
rs133262280	*PODXL2*	532521	U 202.2	Podocalyxin-like 2
	*MCM2*	510120	U 177.6	Minichromosome maintenance complex component 2
	*TPRA1*	617772	U 160.1	Transmembrane protein adipocyte-associated 1
	*LOC10105309*	109905309	U 57.8	Uncharacterized LOC101905309
	*PLXNA1*	531240	D 192.2	Plexin A1
	*CHCHD6*	615934	D 200.9	Coiled-coil helix coiled-coil helix domain-containing 6

^1^ rs136155567: gene in ±600 kb; ^2^ ID = identification; ^3^ D = downstream; U = upstream.

**Table 5 genes-12-01666-t005:** Genes close to the SNPs associated to weaning weight of Braunvieh cattle.

SNP_BTA	Gene in ±250 kb ^1^	Gene ID ^2^	Distance, ^3^ kb	Description
rs43668789	GALM	616676	U 217.4	Galactose mutarotase
	SRSF7	507066	U 201.6	Serine and arginine rich splicing factor 7
	GEMIN6	525263	U 160.6	Gem nuclear organelle associated protein 6
	LOC107132913	107132913	U 156.0	Uncharacterized LOC107132913
	DHX57	540993	U 86.1	Dexh-box helicase 57
	MORN2	616607	U 77.8	MORN repeat containing 2
	ARHGEF33	100335703	Cover	Rho guanine nucleotide exchange factor 33
	SOS1	537682	D 17.0	SOS Ras/Rac guanine nucleotide exchange factor 1
	MIR2284Z-2	102465308	D 62.5	Microrna 2284z-2
	LOC104973309	104973309	D 121.0	Ubiquitin-40S ribosomal protein S27a pseudogene
	CDKL4	517478	D 207.4	Cyclin dependent kinase like 4
	LOC782845	782845	D 241.7	60S ribosomal protein L23a pseudogen
rs136155567_27	LOC104976093	104976093	D 470.9	Uncharacterized LOC104976093
	NRG1	281361	D 567.1	Neuregulin 1

^1^ rs136155567: gene in ±600 kb; ^2^ ID = identification; ^3^ D = downstream; U = upstream.

## Data Availability

Data could be made available upon reasonable request to the authors.

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
