# Peer review of "Discovering of Genomic Variations Associated to Growth Traits by GWAS in Braunvieh Cattle"

_genes, 2021, doi:10.3390/genes12111666_

Round 1

Reviewer 1 Report

Overall, this is an interesting manuscript dealing with a novel topic. To my knowledge, there has not been many studies addressing genetic architecture of growth traits in Braunvieh cattle,  especially not in population outside of Swizerland.. The paper is well structured and written, the methods of data analysis are scientifically sound, the results, discussion, and conclusions are clearly presented.

My suggestions for improvement are given below.

1) Please use acronyms for traits consistently: in some parts of the manuscript BW and WW are used, in some other there are BWT, WWT, and YWT.

2) Would it be possible to provide a PCA graph to see the population stratification?

3) My understanding is that no significant association was found for trait YWT. If this is correct, please state it clearly. Also, I am a little surprised by that result, because the three growth traits should be correlated and thus, some level of either linkage or pleiotropy would be expected among the SNPs influencing these traits. Would it be possible to provide more details/explanation why none of the SNPs was significant for YWT.

Author Response

Response to the reviewer comments

Reviewer 1.

Overall, this is an interesting manuscript dealing with a novel topic. To my knowledge, there has not been many studies addressing genetic architecture of growth traits in Braunvieh cattle,  especially not in population outside of Swizerland.. The paper is well structured and written, the methods of data analysis are scientifically sound, the results, discussion, and conclusions are clearly presented.

Thank you for your comments. All suggestions were considered and implemented in the manuscript. Changes are highlighted along the body of manuscript.

My suggestions for improvement are given below.

1) Please use acronyms for traits consistently: in some parts of the manuscript BW and WW are used, in some other there are BWT, WWT, and YWT.

  1. The acronyms were revised and corrected.

2) Would it be possible to provide a PCA graph to see the population stratification?

  1. A population stratification plot was included.

3) My understanding is that no significant association was found for trait YWT. If this is correct, please state it clearly. Also, I am a little surprised by that result, because the three growth traits should be correlated and thus, some level of either linkage or pleiotropy would be expected among the SNPs influencing these traits. Would it be possible to provide more details/explanation why none of the SNPs was significant for YWT.

R.This kind of outcomes are more common in GWAS analysis. In fact none of the revised literature papers shows SNPs association suggesting pleiotropic traits. This might be related to the effect size on the traits, and We justify the lack of association for YW with the amount of data reduction from BW and WW to YW. This assumption was included as a brief discussion.

Reviewer 2 Report

The paper deals with the growth traits in cattle. It is the GWAS of average quality, but brings some new knowledge. After some revisions could be suitable for publication.

Firstly, the title is a bit overstated. Entitle it more soberly, e.g. “GWAS of growth traits in Braunvieh cattle” or so.

How reliable are the data on the weights, esp. the birth weights. Were the animals kept in common farms or in breeding companies?

Methodical comments. The time span is rather long, and the animals were raised under different conditions (chapter 2.1). It could influence the results of the association study. Please, discuss it.

You have detected only three SNPs with significant association to growth. Compare with other association studies, is the number usual, or different numbers of SNPs are detected by other authors?

The arrangement of tables, figures and text in pages 4-5 is labyrinthine. Rearrange it, divide the tables and figures by the text, e.g.

I do not understand fully the Figure 2. You analyze three traits, in the legend are BW, WW and YW, but there are only two figures without designation?

Chapter Discussion, paragraph “Two SNPs markers…. (Table 5).” only repeats the results from the Table 5. Delete it or rewrite.

The same chapter, the last paragraph. You write on the location of one SNP in the intronic region of the gene. It can hint at the association to the respective gene. But, in the introns are coded also interfering RNAs. Have you analyzed also the possible effect of those? The data on the RNAi and growth from other authors?

Formal imperfections

The rows are not numbered, which complicates the orientation

Check throughout the text the writing of gene names in italic. In Abstract GALM, in Tables 4 and 5, etc.

Author Response

Response to the reviewer comments

Reviewer 2.

Thank you for your comments. All suggestions were considered and implemented in the manuscript. Changes are highlighted along the body of manuscript.

The paper deals with the growth traits in cattle. It is the GWAS of average quality, but brings some new knowledge. After some revisions could be suitable for publication.

Firstly, the title is a bit overstated. Entitle it more soberly, e.g. “GWAS of growth traits in Braunvieh cattle” or so.

 R: Title was modified as recommended.

How reliable are the data on the weights, esp. the birth weights. Were the animals kept in common farms or in breeding companies?

R: As described in manuscript, animals are kept in different herds (ranches). Since animals from these ranches are registered in the Mexican Braunvieh Cattle Association, the recording of data is mandatory and weekly collected by a Technical committee, so they are reliable.

Methodical comments. The time span is rather long, and the animals were raised under different conditions (chapter 2.1). It could influence the results of the association study. Please, discuss it.

  1. Statistical model included year effect.

You have detected only three SNPs with significant association to growth. Compare with other association studies, is the number usual, or different numbers of SNPs are detected by other authors?

R: For discussion an extensive searching for association studies was made. In general, different SNPs are reported (this is known by the different arrays used), yet some assessments bring out some similar results on QTLs and Genes related.  In our case, none similar findings were found, but purported effects on associated regions, QTLs and SNPs were broadly disscused.

The arrangement of tables, figures and text in pages 4-5 is labyrinthine. Rearrange it, divide the tables and figures by the text, e.g.

 R: The tables and figures were arranged as close as possible to the location of firts reference in text. Since Figure 1 was included, the numbers were  also modified.

I do not understand fully the Figure 2. You analyze three traits, in the legend are BW, WW and YW, but there are only two figures without designation?

 R: Yearling weight graph was missing somehow. The plot was included in this version of the manuscript.

Chapter Discussion, paragraph “Two SNPs markers…. (Table 5).” only repeats the results from the Table 5. Delete it or rewrite.

R: The paragraph was modified as suggested.

The same chapter, the last paragraph. You write on the location of one SNP in the intronic region of the gene. It can hint at the association to the respective gene. But, in the introns are coded also interfering RNAs. Have you analyzed also the possible effect of those? The data on the RNAi and growth from other authors?

R: A discussion on this observation was included as suggested, to properly conclude.

Formal imperfections

The rows are not numbered, which complicates the orientation

R: We follow the recommendations and used the manuscript template without lines. We are sorry for that omission.

Check throughout the text the writing of gene names in italic. In Abstract GALM, in Tables 4 and 5, etc.

R. Names were checked and corrected.
